# The Role of CTLA-4 in T Cell Exhaustion in Chronic Hepatitis B Virus Infection

**DOI:** 10.3390/v15051141

**Published:** 2023-05-10

**Authors:** Ása Didriksen Apol, Anni Assing Winckelmann, Rasmus Bülow Duus, Jens Bukh, Nina Weis

**Affiliations:** 1Department of Infectious Diseases, Copenhagen University Hospital, 2650 Hvidovre, Denmark; asa.didriksen.apol.01@regionh.dk (Á.D.A.); anni.assing.winckelmann@regionh.dk (A.A.W.); rasmus_bulow_duus@hotmail.com (R.B.D.); jbukh@sund.ku.dk (J.B.); 2Department of Clinical Medicine, Faculty of Health and Medical Sciences, University of Copenhagen, 2200 Copenhagen, Denmark; 3Copenhagen Hepatitis C Program (CO-HEP), Department of Infectious Diseases, Copenhagen University Hospital, 2650 Hvidovre, Denmark; 4Copenhagen Hepatitis C Program (CO-HEP), Department of Immunology and Microbiology, Faculty of Health and Medical Sciences, University of Copenhagen, 2200 Copenhagen, Denmark

**Keywords:** T cell exhaustion, CTLA-4, chronic hepatitis B, immune checkpoint inhibitors

## Abstract

Patients with chronic hepatitis B (CHB) gradually develop T cell exhaustion, and the inhibitory receptor molecule, cytotoxic T-lymphocyte antigen-4 (CTLA-4), may play a role in this phenomenon. This systematic review investigates the role of CTLA-4 in the development of T cell exhaustion in CHB. A systematic literature search was conducted on PubMed and Embase on 31 March 2023 to identify relevant studies. Fifteen studies were included in this review. A majority of the studies investigating CD8^+^ T cells demonstrated increased expression of CTLA-4 in CHB patients, though one study found this only in HBeAg-positive patients. Three out of four studies investigating the expression of CTLA-4 on CD4^+^ T cells found upregulation of CTLA-4. Several studies showed constitutive expression of CLTA-4 on CD4^+^ regulatory T cells. CTLA-4 blockade resulted in heterogeneous responses for all T cell types, as it resulted in increased T cell proliferation and/or cytokine production in some studies, while other studies found this only when combining blockade of CTLA-4 with other inhibitory receptors. Although mounting evidence supports a role of CTLA-4 in T cell exhaustion, there is still insufficient documentation to describe the expression and exact role of CTLA-4 in T cell exhaustion in CHB.

## 1. Introduction

Hepatitis B virus (HBV) infection is a major global public health threat. In 2019, an estimated 296 million people were living with chronic hepatitis B (CHB). CHB can lead to liver cirrhosis and hepatocellular carcinoma (HCC) and is estimated to have caused 820,000 deaths in 2019. HBV infection can either result in an acute infection, which is cleared by the immune system, or CHB, defined as the presence of the HB antigen in the blood for more than six months [1].

HBV is a partially double-stranded DNA virus that primarily infects hepatocytes. The currently available prophylactic hepatitis B vaccine effectively prevents infection with HBV, but the vaccine has no effect on established chronic infection [2]. The currently available treatments for CHB are often effective in suppressing viral replication. However, due to the long-term persistence of viral covalently closed circular DNA (cccDNA) in the hepatocytes, treatment is not curative and is often life-long [3,4,5]. Therefore, there is a need for novel therapeutic approaches for the treatment of CHB.

HBV is mainly non-cytopathic, and the host immune response to the virus plays a crucial role in liver injury and virus control. A dynamic balance between immune tolerance and immune clearance accounts for the outcome in patients with CHB [6]. Virus-specific CD8^+^ T cells recognize viral antigens present on infected hepatocytes with help from CD4^+^ T cells. However, this CD8^+^ T cell response is often weak or undetectable in patients with CHB; over time, patients with CHB develop a permanent weakening of the T cell-dependent immune system. This results in an inadequate or dysfunctional T cell response to HBV, a phenomenon known as exhaustion [7].

In recent years, the underlying mechanisms of T cell exhaustion in patients with CHB have been increasingly investigated. It has been established that T cell exhaustion is especially characterized by reduced cytokine production, poor effector cytotoxic activity, and sustained expression of multiple inhibitory receptor molecules, including cytotoxic T-lymphocyte antigen-4 (CTLA-4) [8]. CTLA-4 is one of multiple inhibitory molecules which can reduce T cell responses through co-stimulation and inhibitory signals. This is a vital mechanism to prevent immune-mediated tissue damage; however, several studies have described that overexpression of CTLA-4 in CHB patients can be linked to immune dysfunction leading to viral persistence [7,9,10,11]. Blocking this inhibitory receptor could therefore restore the ability of the antiviral T cells to clear HBV. Since anti-CTLA-4 drugs are currently available for cancer immunotherapy, this might be a promising approach for future CHB treatment [4]. Hence, this current review aims to investigate which role the CTLA-4 receptor plays in the exhaustion state of T cells in patients with CHB.

### 1.1. The Immune System and Hepatitis B Virus

The natural history of chronic HBV infection is a dynamic and complex process. The disease progression is not linear, the infection progresses through several recognizable phases, and the complexity of the immunologic response has shown considerable variation [7,12]. The T cell responses are known to play a significant role in the control of HBV infection and liver inflammation. Thus, they are important determinants of disease progression and clinical immune status of the patients [7].

#### 1.1.1. The T Cell-Mediated Immune Response

The T cell-mediated immune response plays a significant role in viral clearance during HBV infection [8,13]. Antigen-specific CD8^+^ T cells are essential in the immune control of a chronic HBV infection. CD8^+^ T cells directly eliminate virus-infected hepatocytes by cytolytic and non-cytolytic mechanisms and often require help from CD4^+^ T cells [14]. CD4^+^ T cells play a complex role in the regulation of the body’s immune response during a viral infection. In part, they play a key role in activating and maintaining effective cytolytic CD8^+^ T cells and the differentiation to various T cell subsets, for example, follicular helper cells (Tfh), T helper 1 (Th1), T helper 2 (Th2), and T regulatory cells (Tregs), are essential for effective viral control and regulation [7,8]. Finally, the subgroup of CD4^+^ T cells, Tregs, play an important role in regulating the HBV-specific immune response. CD4^+^ Tregs are a subpopulation of CD4^+^ T cells characterized by the constitutive expression of CD25^+^, the transcription factor scurfin, encoded by the forkhead family transcription factor 3 (FoxP3) gene, as well as CTLA-4 [15]. Tregs can act by suppressing the functions of effector T cells. Therefore, they may be a key player in the impaired immune response in CHB, thus influencing disease progression [16,17].

During HBV infection, antigen-presenting cells (APC) process virus-specific antigens and present them on major histocompatibility complexes I and II (MHC-I and MHC-II), respectively [13]. These antigens are subsequently recognized by specific T cell receptors (TCR) on CD8^+^ and CD4^+^ T cell subsets, and the T cells are activated through co-stimulatory CD28 activation [7]. Whether the T cells are activated or inhibited depends on a balance between stimulatory and inhibitory signals. In addition to MHC-TCR binding, successful T cell activation requires the T cell CD28 receptor to bind to either CD80 or CD86, expressed on the surface of the APCs. CD28 is expressed on the surface of CD4^+^ and CD8^+^ T cells and is the primary co-stimulatory molecule in T cell activation, expansion, and survival. CTLA-4 is a receptor molecule on the surface of T cells and is particularly abundant on CD4^+^ Tregs. CTLA-4 competitively binds to CD80 and CD86, thus inhibiting the positive co-stimulatory signal of CD28, resulting in a strong inhibitory signal in the T cells (see Figure 1). The balance and competition between these co-receptors have proven to be important for our understanding of T cell exhaustion and may contribute to our understanding of the immunopathogenesis of CHB [7].

#### 1.1.2. Cytotoxic T-Lymphocyte Antigen-4 (CTLA-4)

CTLA-4, also known as CD152, belongs to the immunoglobulin superfamily and is a receptor molecule located in the membrane of both CD4^+^ and CD8^+^ T cells [18]. The CTLA-4 gene is placed on band q33 of chromosome 2 and encodes the 223 amino acid CTLA-4 protein, consisting of an extracellular V-domain, a transmembrane domain, and a cytoplasmatic tail [18]. The CTLA-4 gene and the CD28 gene exhibit extensive homology at the nucleotide level; however, CTLA-4 binds to CD80/CD86 with an affinity 20-fold higher than CD28 [19]. CTLA-4 does not appear to be expressed on resting T cells but is upregulated following activation and progression into the cell cycle, which is thought to have a stabilizing function on the immune system [18]. CTLA-4 is considered an immune checkpoint molecule, and the general function of these molecules is to regulate the magnitude of an immune response, protect against immune-mediated tissue damage during infection, and maintain self-tolerance [4].

#### 1.1.3. CTLA-4 Inhibitory Mechanisms

The specific inhibitory mechanisms related to CTLA-4 are complex, and several possible mechanisms by which CTLA-4 may inhibit T cell responses have been proposed. As mentioned above, CTLA-4 inhibits the positive co-stimulatory signaling of CD28 by competitively binding to CD80 and CD86 [7]. There are indications that the cytoplasmic tail of CTLA-4 has an intrinsic inhibitory function, which through a cascade of intracellular reactions can inhibit the CD28/TCR signal, thus leading to inhibition. Previous studies have demonstrated that CTLA-4 can activate the idolenamin2-3 deoxygenase (IDO) pathway in the APCs. The IDO enzyme breaks down tryptophan, an amino acid essential for T cell proliferation, and the reduced amount of tryptophan eventually leads to T cell exhaustion. Lastly, it has been shown that CD4^+^ Tregs constitutively express CTLA-4, which is unusual for non-activated T cells. Tregs have a regulatory and suppressor function, and several studies suggest a role for CTLA-4 in the function of Tregs [18].

## 2. Materials and Methods

This systematic review was performed by the Preferred Reporting Items for Systematic Reviews and Meta-Analysis (PRISMA) 2020 reporting guidelines [20]. Key Population Intervention Comparison Outcome (PICO) questions were identified and developed (see Table 1).

### 2.1. Eligibility Criteria

All types of studies were eligible for inclusion except reviews and meta-analyses. The inclusion criteria for studies investigating CTLA-4 on T cells in CHB patients were (all required) as follows:Adult study population ≥18 years;Comparison to controls (healthy or other);Studies in English language.

The exclusion criteria were as follows:Age < 18 years;Studies performed on animals;Review or meta-analysis;Only a subgroup of CHB clinical stage initially included in study.

The names of the authors or journals of the articles did not influence the decision to exclude or include the articles.

### 2.2. Search Strategy

A comprehensive systematic literature search was conducted of PubMed and Embase on 31 March 2023. Only studies in English language were included. The primary search string was formed using a search table with a simple block structure consisting of two blocks, CTLA-4 and hepatitis B, respectively (see Appendix A). A controlled (MeSH Terms (in PubMed) and Emtree (Embase)) and an uncontrolled (Text Word) search were conducted in each block, after which these were combined. Subsequently, the two blocks were combined.

### 2.3. Study Selections

Titles and abstracts were screened for potential eligibility. Full texts of the included abstracts were retrieved and screened. DistillerSR software was used for the screening process. The records were screened by one reviewer (Á.D.A.). 

### 2.4. Statistical Analysis

Descriptive statistics were performed to summarize the literature findings by using Microsoft Excel (Version 2202 Build 16.0.14931.20960).

## 3. Results

### 3.1. Search Results

The electronic literature search resulted in 521 articles after duplicates were removed. The sorting process was divided into three phases. Based on the title and/or abstract, 450 articles were excluded based on relevance, accessibility, or language. Of the remaining 71 articles, 15 were included as research articles after full-text screening, while some of the excluded articles were used for background information on the topic. Figure 2 illustrates the flow chart of article inclusion.

All articles were published between 2005 and 2023. Nine articles were from China, two each from the Netherlands and Germany, and one each from England and the United States. Twelve of the fifteen included studies were cross-sectional, two were descriptive, and one was a combined cross-sectional and longitudinal study. Table 2 shows baseline characteristics for the included studies.

### 3.2. Patient Characteristics

All included patients were chronically infected with HBV. The exclusion criteria of the included studies were similar, as most studies excluded patients coinfected with other hepatitis viruses and/or human immunodeficiency virus (HIV) [11,17,21,22,23,24,25,26,27,29,30,31], with concurrent liver disease [16,21,22,23,24,26,29,30], and/or patients receiving antiviral or immunomodulatory therapy [11,15,16,17,21,23,24,28,30]. One study did not provide detailed information on patients’ inclusion or exclusion criteria [9].

This review included 1192 CHB patients and 218 healthy controls. Several studies also included additional subgroups, such as acute HBV infection (AHB) or patients with CHB divided into subgroups based on different clinical stages of the disease. The mean age of CHB patients ranged from 33 to 46 years, and male subjects dominated the patient population (in studies where information about the sex of the included patients was available). Appendix A show an overview of the findings in the included studies, where the role of CTLA-4 in T cell exhaustion was examined in relation to different types of T cells, divided into CD8^+^, CD4^+^, and Tregs, respectively.

### 3.3. CTLA-4 and CD8^+^ T Cell Exhaustion

The studies investigating CTLA-4 expression on CD8^+^ T cells in CHB patients found varying results, but the general trend pointed towards upregulation of CTLA-4 on CD8^+^ T cells—one study demonstrated this only in HBeAg-positive CHB patients [24].

A study by Wang, X et al. [9] investigated the differences in the expression of exhaustion markers on intrahepatic and peripheral blood CD8^+^ T cells from 40 CHB patients with HCC. Liver samples were taken from tumor (HCC) and para-tumorous (CHB) tissue and peripheral blood mononuclear cells (PBMCs) were obtained from the same patients and 40 healthy controls. The study found significantly higher expression levels of all exhaustion markers, including CTLA-4, in both CHB and HCC liver tissue compared to the corresponding patients’ PBMCs. The study also demonstrated that the functional states of CD8^+^ T cells in both CHB and HCC were compromised, as evidenced by decreased proliferation and cell activity and reduced production of cytokines compared to corresponding patients’ PBMCs and healthy controls. Further, compared to CD8^+^ T cells from CHB liver tissue, CD8^+^ T cells from HCC tissue generally showed higher expression of exhaustion markers, lower proliferation levels, and lower cell activity [9].

Tang et al. [28] investigated the correlation between CD28 family receptors (the CD28 family consists of the co-stimulatory receptors, CD28 and inducible T cell co-stimulator (ICOS), and three co-inhibitory receptors, PD-1, CTLA-4, and B- and T-lymphocyte attenuator (BTLA) [28]), including CTLA-4, on T cells and the development of exhaustion in the peripheral blood of 52 patients with CHB compared to 26 healthy controls. The study did not find any upregulation of CTLA-4 expression on CD8^+^ T cells in CHB patients compared to healthy controls, and neither did it demonstrate any correlation between increased viral load and increased CTLA-4 expression on CD8^+^ T cells. However, the study demonstrated a significant positive correlation between viral load and programmed cell death protein 1(PD-1) expression levels on the CD8^+^ T cell membrane [28].

Bengsch et al. [21] studied CHB patients with HBV-specific CD8^+^ T cells and primarily detected an HBV-specific T cell response in HBeAg-negative patients. Accordingly, the authors observed a clear hierarchy of inhibitory receptor expression in PBMCs dominated by PD-1 on HBV-specific CD8^+^ T cells (79.3%), and the expression of CTLA-4 was significantly weaker (21.4%). CTLA-4 blockade resulted in an increase in T cell proliferation; however, PD-1 blockade was far more effective [21].

Schurich et al. [27] examined the contribution of CTLA-4 to CD8^+^ T cell tolerance in 86 CHB patients, 3 patients with resolved HBV infection, and 23 healthy controls. The study investigated both peripheral blood and intrahepatic cells and showed that global CD8^+^ T cells (i.e., non-HBV-specific) in patients with CHB had an increased expression of CTLA-4, which correlated to viral load. The authors found that HBV-specific CD8^+^ T cells expressed more CTLA-4 on the surface compared to global CD8^+^ T cells. Further, they demonstrated that CTLA-4 is upregulated on those HBV-specific CD8^+^ T cells with the highest levels of the proapoptotic protein Bim. Bim is a part of the Blc-3 family, and through activation of Bax, it can induce apoptosis via the mitochondrial pathway. The authors showed that CTLA-4 blockage led to reduced Bim expression [27].

A study by Peng et al. [24] (2011) investigated the correlation between HBeAg and the properties of HBV-specific CD8^+^ T cells in the PBMCs of 103 CHB patients and 30 healthy controls. No significant differences in HBV-pentamer CD8^+^ T cell frequency were found between HBeAg-positive and HBeAg-negative CHB patients, but increased PD-1 and CTLA-4 expression on HBV-specific CD8^+^ T cells was seen in the HBeAg-positive group. HBV peptide stimulation with anti-PD-L1 (programmed cell death ligand 1; the ligand to PD-1) and anti-CTLA-4 significantly increased the proliferation of PBMCs from both groups but enhanced IFN-γ production only in the HBeAg-positive patients. No functional restoration was observed when cells were treated with anti-PD1 or anti-CTLA-4 alone [24].

Jiang et al. [22] investigated the phenotypic heterogeneity of exhausted CD8^+^ T cells in the peripheral blood of 31 CHB patients and 23 healthy controls. Compared to healthy controls, CD8^+^ T cells in CHB patients had higher expression levels of the inhibitory receptors lymphocyte-activation gene 3 (LAG3) and T cell immunoglobulin and mucin domain-containing protein 3 (TIM3). In contrast, the increased expression levels of CTLA-4 and PD-1 were non-significant. To further identify the CD8^+^ T cell subset expressing inhibitory receptors, the authors studied a subset of CD8^+^ T cells expressing C-X-C motif chemokine receptor type 5 (CXCR5), which is known to direct migration to B-cell follicles. They found a higher expression of inhibitory receptors, including CTLA-4, on the surface of CXCR5^+^CD8^+^ T compared to CXCR5^−^CD8^+^ T cells, but only PD1 and TIM3 were significantly elevated. In CHB patients, the subset of follicular cytotoxic T (Tfc) cells expressed higher levels of TIM3 and PD1, but not CTLA-4, and the cytotoxic T cell (Tc) 17 subset displayed significantly increased expression of only TIM3 and LAG3 in CHB patients [22].

In a study of 121 CHB patients, Yu et al. [11] investigated whether RNA interference (RNAi) could block CTLA-4 in human lymphocytes from peripheral blood in vitro and promote the secretion of proinflammatory cytokines, such as IFN-γ and IL-2. The study demonstrated that CTLA-4-targeted RNAi significantly suppressed the expression of human CTLA-4 mRNA and could induce a non-cytolytic Th1/Th2 response [11].

### 3.4. CTLA-4 Is Upregulated on CD4^+^ T Cells

Four out of five included studies investigating the expression of CTLA-4 on CD4^+^ T cells found upregulation of CTLA-4.

A study by Raziorrouh et al. [25] investigated the expression of multiple inhibitory molecules, including CTLA-4, in peripheral blood of 66 CHB patients, as well as 41 patients with AHB, 5 HBV resolvers, and 7 healthy controls. In 30 CHB patients, the HBV-specific CD4^+^ T cells were dominated by PD-1 expression (77.9%) compared to CTLA-4 (19.4%). Incubation of the PBMCs with anti-CTLA-4 and anti-PD-L1/L2 demonstrated that PD-1 blockade restituted CD4^+^ T cell proliferation, whereas CTLA-4 blockade failed to increase the same cells’ ability to proliferate [25]. These findings are substantiated by Tang et al. [28], whose data indicate only a minimal upregulation of CTLA-4 on peripheral HBV-specific CD4^+^ T cells compared to a group of healthy controls. In contrast, Tang et al. demonstrated that the global CD4^+^ T cells had an increased CTLA-4 expression compared to the healthy individuals [28]. These findings were consistent with a study by Park et al. [23], who also found that CTLA-4 expression levels in peripheral blood CD4^+^ T cells were greater in CHB patients than in uninfected controls [23].

A study by Wang, L. et al. [29] with a total of 62 CHB patients found that the expression levels of CTLA-4 on CD4^+^ T cells in peripheral blood prior to and following treatment with antiviral agents were significantly decreased compared to 30 healthy controls. In addition, the study did not find any correlation between HBV DNA levels and CTLA-4 expression in CD4^+^ T cells. Moreover, compared to the healthy control group, the total levels of CD4^+^ and CD8^+^ T cells were significantly decreased [29].

Wen et al. [26] investigated the role of the T cell subset CD4^+^CXCR5^−^FOXP3^+^ T cells with CTLA-4 expression in a series of CHB patients in both a cross-sectional cohort (106 treatment-naïve CHB patients, 25 healthy controls, and 13 patients with HBV-related hepatic failure) and a longitudinal cohort (15 HBeAg-positive CHB patients treated with telbivudine (antiviral drug)). The study revealed an upregulation of immunosuppressive features, such as CTLA-4 and PD-1, on CD4^+^CXCR5^−^FOXP3^+^ T cells in treatment-naïve CHB patients and in patients with HBV-related hepatic failure compared to healthy controls. Stimulation with HBeAg and HBcAg significantly upregulated the expression of inhibitory markers, such as CTLA-4 and PD-1, within the CD4^+^CXCR5^−^FOXP3^+^ T cell population. In the longitudinal cohort, the frequency of CD4^+^CXCR5^−^FOXP3^+^ T cells was significantly lower in patients with a complete response (i.e., HBeAg seroconversion and HBV DNA levels < 300 copies/mL) than in non-complete responders, and a substantial decrease in CTLA-4 expression at week 12 was observed in complete responders compared to non-complete responders.

### 3.5. Elevated Levels of CTLA-4 Expression on CD4^+^ Tregs

Several of the included studies investigating Tregs showed constitutive expression of CLTA-4 on CD4^+^ regulatory T cells, and moreover, it was demonstrated that the Treg subtype, CD4^+^CD25^+high^, had significantly elevated expression of CTLA-4 [15]. A study demonstrated an intrahepatic Treg subgroup that was completely absent in peripheral blood [31].

A study by Peng et al. [15] investigated circulating Tregs in PBMCs of 79 CHB patients, 26 asymptomatic carriers (ASC), 12 patients with AHB, and 20 healthy controls, respectively. The frequency of the Treg subtype CD4^+^CD25^+high^ (which has high levels of CD25 expression) in HBeAg-positive CHB patients was significantly higher than in healthy controls, which was not the case for HBeAg-negative patients nor ASCs. There was no significant difference in the frequency of the total CD4^+^CD25^+^ T cell population between groups. Compared to CD25^−^ T cells, CD4^+^CD25^+high^ cells had significantly elevated expression of CTLA-4. The frequency of CD4^+^CD25^+high^ was found to be positively correlated with serum viral load, and the Tregs were capable of suppressing proliferation and IFN-γ production of autologous PBMCs in vitro after HBV antigen stimulation. The combined blockade of PD-1 and CTLA-4, using anti-PD-1 and anti-CTLA-4 monoclonal antibodies, slightly enhanced the cellular proliferation and significantly increased IFN-γ production of PBMCs co-cultured with Tregs [15].

In contrast to the abovementioned study, Stoop et al. (2005) [17] detected a higher percentage of Tregs, defined as CD4^+^CD25^+^CD45RO and CTLA-4-positive, within the population of CD4^+^ T cells in the peripheral blood of 50 CHB patients compared to 23 healthy controls and 9 individuals with resolved (i.e., functionally cured) HBV infection. Depletion of CD25^+^ cells from PBMCs of CHB patients resulted in an enhanced proliferation after stimulation with the HBV antigen [17]. These findings were confirmed by Zhang et al. [16], who studied Tregs in 49 HCC patients, 15 CHB patients, and 25 healthy controls. They found both increased circulating and liver-resident Tregs in CHB patients compared to the control group. Moreover, the study showed that when healthy donor PBMCs were co-cultured with human hepatoma cells HepG2.2.15 (transfected with HBV) and its parental cell line HepG2, the Treg population increased and upregulated the expression of CTLA-4 on HepG2.2.15 compared with the HepG2 cells [16].

Another study by Stoop et al. (2008) [31] examined the intrahepatic phenotype of Tregs in a total of 37 CHB patients and found that the liver contained a population of CD4^+^CD25^−^FoxP3^+^ cells that were completely absent in peripheral blood. When studying a subset of seven CHB patients, it was demonstrated that the “conventional” CD4^+^CD25^+^FoxP3^+^ Treg consistently expressed higher levels of CTLA-4 and HLA-DR but not PD-1 compared to CD4^+^CD25^−^FoxP3^+^ cells.

The studies of Stoop et al. (2005) [17] and Park et al. [23] did not find any correlation between viral load, serum ALT levels, and the frequency of circulating Tregs in CHB patients. Similar findings were made in Stoop et al.’s (2008) [31] other study, where no correlation was found between serum ALT levels or Metavir score and the proportion of intrahepatic Tregs. However, patients with a high viral load were found to have a higher proportion of Tregs in the liver but not in the blood.

## 4. Discussion

Despite several years of research in the field of HBV, it is not yet clear which factors contribute to the progression of CHB nor the reason for the impaired virus-specific T cell responses in these patients. This systematic review suggests that the CTLA-4 receptor plays an important role.

### 4.1. CD8^+^ and CD4^+^ T Cells

In the current review, several studies demonstrate an upregulation of CTLA-4 on both CD8^+^ and CD4^+^ T cells in CHB patients [9,23,27,28]. However, other inhibitory receptors, such as PD-1, were also over-represented [9,22,25,27,28], suggesting that PD-1 contributes to a greater extent to HBV-specific dysfunction. Nevertheless, it is worth noting that expression of CTLA-4 is described to be activation-dependent [18], and one study showed that CTLA-4 was indeed upregulated upon stimulation of HBV-specific CD8^+^ T cells with the HBV antigen [27]. Therefore, particularly in activated T cells, CTLA-4 could indeed have a significant role in the immune exhaustion in CHB. These findings point towards the possibility that CTLA-4 blockade could increase CD8^+^ T cell function if the receptors are upregulated after antigen-specific stimulation. In addition to CTLA-4 and PD-1, other inhibitory receptors, such as LAG3, TIM3, and BTLA, also hold interesting research perspectives. Some studies included in this review did in fact demonstrate upregulation of these receptors on T cells of CHB patients [22,28]. Further investigation of these receptors is warranted but is beyond the scope of this review.

Inhibition of the CTLA-4 signaling pathways by monoclonal antibodies is currently an available treatment in oncologic patients [32], and therefore, CTLA-4 blockade could be a promising approach for the treatment of CHB. Receptor blockade directed at several different inhibitory receptors resulted in heterogeneous responses in the included studies. Several studies failed to demonstrate a relevant increase in CD8^+^ T cell proliferation after CTLA-4 blockade, whereas PD-1 blockade did lead to increased expansion of T cell responses in CHB patients [21,27]. The lack of T cell reconstitution after CTLA-4 blockade may reflect the role of other inhibitory receptors, such as PD-1. It is also possible that the expression of CTLA-4 on other T cell subtypes, such as Tregs, could counteract the effect of CTLA-4 blockade on CD8^+^ T cells. Interestingly, in one study the combined blockade of CTLA-4 and PD-1 led to increased CD8^+^ T cell proliferation and enhanced IFN-γ production in HBeAg-positive patients, whereas no such response was observed when blocking either PD-1 or CTLA-4 alone [24]. One could speculate that these conflicting results are due to differences in the initial expression levels of inhibitory receptors on the T cells of the CHB patients in different clinical stages, and therefore, a different response to blockade. These observations might also be explained by the synergy of multiple inhibitory receptors on T cells and that T cell exhaustion is regulated by a complex network of various co-expressed inhibitory receptors. These findings may have important therapeutical implications, and the heterogeneity of the responses indicates that potential future candidates for immune checkpoint therapy must be carefully selected. The fact that only HBeAg-positive patients were found to have an upregulation of CTLA-4 and an increased cytokine production after combined CTLA-4 and PD-1 blockade suggests that different stages of CHB could have diverging effects of immune therapy.

High viral replication levels in CHB patients reflect an impaired immune control of the virus. Thus, the exploration of the association between viral load and CTLA-4 expression in several of the included studies is interesting. The positive correlation between increased expression of CTLA-4 on CD8^+^ T cells and viral load demonstrated in some of the studies [9,27] indicates an association between increased viral replication and incipient upregulation of CTLA-4. This suggests that CTLA-4 plays an important role in the development of CD8^+^ T cell dysfunction. Contrary to these findings, other studies did show a positive correlation between increased expression of PD-1 on CD8^+^ T cells and viral load but failed to demonstrate this correlation for CTLA-4 [21,28]. These results indicate that PD-1 is important in the development of CD8^+^ T cell dysfunction, while the role of CTLA-4 is less significant.

The positive correlation between CTLA-4 and Bim demonstrated in one included study [27] suggests that CTLA-4 signaling could help to drive a proapoptotic phenotype and contribute to increased apoptosis in HBV-specific CD8^+^ T cells, thus leading to exhaustion of these cells. The fact that CTLA-4 blockade led to reduced Bim expression levels in HBV-specific CD8^+^ T cells indicates a potential role for CTLA-4 in driving a Bim-mediated weakening of the antiviral response. Two studies [9,11] suggest that CTLA-4 can inhibit the secretion of proinflammatory cytokines from activated CD8^+^ T cells, leading to weakening of the non-cytolytic function of CD8^+^ T cells. This implies that CTLA-4 is important in regulating or suppressing the non-cytolytic Th1/Th2 response in CD8^+^ T cells.

Theoretically, the suppression of viral replication during antiviral treatment could induce a reversion of immune exhaustion due to the reduction in HBV antigen exposure of the immune cells. However, this review does not substantiate this hypothesis regarding CD4^+^ T cells, as a study found no significant differences in CTLA-4 expression levels on CD4^+^ T cells prior to and following antiviral treatment [29]. The fact that stimulation with HBV antigens in a different study led to an upregulation of both CTLA-4 and PD-1 on CD4^+^CXCR5^−^FOXP3^+^ T cells [26] implies that HBV antigens enhance the immunosuppressive features of these cells in CHB. An interesting observation in this study was the fact that the frequency of CTLA^+^CD4^+^CXCR5^−^FOXP3^+^ T cells was associated with a virological response in CHB patients undergoing antiviral treatment, suggesting that CTLA-4 expression on these cells is related to an unfavorable outcome in CHB patients receiving antiviral therapy. This provides interesting perspectives, implicating that targeting CTLA^+^CD4^+^CXCR5^−^FOXP3^+^ T cells might be a novel therapeutic approach for HBV treatment.

### 4.2. Tregs

The included studies found constitutively upregulated CTLA-4 expression on Tregs in CHB patients. Since Tregs are believed to contribute to the exhausted immune state in CHB, the percentage of Tregs relative to total CD4^+^ T cells could serve as an indicator of the degree of immune exhaustion. The proportion of Tregs in CHB patients in the included studies ranged from increased [16,17] to unchanged [15] compared to healthy controls. However, the latter study [15] showed that the specific CD4^+^CD25^+high^ Treg subtype was significantly increased in HBeAg-positive patients compared to healthy controls and showed a positive correlation between CD4^+^CD25^+high^ and viral load [15]. CD4^+^CD25^+high^ Tregs generally have a high expression of CTLA-4, and these results indicate that CD4^+^CD25^+high^ Tregs are upregulated with increasing HBV replication. This is substantiated by the fact that both the HBV DNA and the CD4^+^CD25^+high^ Treg levels were reduced after antiviral therapy. The antiviral therapy contributed to the inhibition of HBV replication, and the reduction in viral load may induce a downregulation of CTLA-4 potent CD4^+^CD25^+high^ Tregs. These observations indicate that over time, CD4^+^CD25^+high^ Tregs may contribute to T cell dysfunction and that effective treatment may prevent this dysregulation. On the other hand, the fact that other studies failed to show any correlation between viral load and the percentage of Tregs in peripheral blood [17,23] somewhat contradicts the above-mentioned hypothesis. Similarly, another study did not find a correlation between a high viral load and the percentage of Tregs in peripheral blood of CHB patients [31], but interestingly, they did find this in the liver. Since Tregs play a role in regulating the immune response [16,17], the liver damage caused by HBV potentially is a stimulus for the induction of Tregs. As the liver is the primary site of HBV infection, it is more likely that the immune system is altered in the intrahepatic compartment, hence the higher proportion of Tregs in the liver compared to the blood. On the contrary, the lack of correlation between the proportion of intrahepatic Tregs and either the Metavir score or ALT levels challenges this hypothesis. The fact that Stoop et al. [31] found an intrahepatic immune cell type that is completely absent in peripheral blood further substantiates the need for future studies to uncover this field.

The included studies on T cell exhaustion in HBV infection make it clear that the pool of exhausted T cells consists of phenotypically and functionally different T cell subsets with diverse levels of responsiveness to intervention. A possible explanation for the differing results of the included studies might be that a subpopulation within a T cell subset (e.g., CD4^+^CXCR5^−^FOXP3^+^ T cells or CD4^+^CD25^+high^) is actually the driving force of T cell exhaustion in HBV infection, which some of the studies indicate [15,22,26]. This highlights the possibility that a subset of either CD8^+^ or CD4^+^ T cells can provide a future immunotherapeutic target in the treatment of CHB.

### 4.3. CTLA-4 in a Clinical Setting

The fact that CTLA-4 blockade resulted in quite heterogeneous T cell responses in these ex vivo studies using blood and/or liver tissue collected from individuals with CHB raises the question of whether this approach can be used therapeutically in CHB patients. CTLA-4 inhibitors, such as ipilimumab or tremelimumab, are currently used for cancer immunotherapy and have shown a survival benefit and/or disease control in the treatment of several advanced cancers [32]. However, a known adverse effect of CTLA-4 blockade is immune-mediated hepatotoxicity. This has led to concerns regarding CTLA-4 blockade in patients with concomitant HBV infection, who may have impaired hepatic function at baseline. Due to substantial theoretical risks, including infiltration of reinvigorated T cells into the liver resulting in liver inflammation, most trials with CTLA-4 inhibition have excluded patients with CHB [10,32]. A case series describing nine patients with melanoma treated with ipilimumab with concomitant HBV or hepatitis C virus (HCV) infection (five HBV, four HCV) demonstrated that the rate of hepatotoxicity in these patients appeared similar to what was seen in the general population. In six patients, viral titers dropped. Interestingly, in two patients who did not develop progressive melanoma (one HBV, one HCV) and therefore presumably responded to ipilimumab, viral titers decreased significantly, suggesting a positive effect [33]. In a recent systematic review investigating the safety and efficacy of immune checkpoint inhibitors (PD-1 and/or CTLA-4 inhibitors) in HBV-/HCV-infected cancer patients, the authors concluded that immune checkpoint inhibitors are considered safe and effective in advanced cancer patients with HBV/HCV infection. Nevertheless, due to the risk of reactivating the hepatitis virus, the authors recommend that patients with viral hepatitis must be monitored closely and treated with antiviral therapy if indicated before or during treatment with immune checkpoint inhibitors [34].

### 4.4. Strengths and Limitations

The majority of the studies included in this review had similar inclusion and exclusion criteria with general patient similarity, where especially competing illnesses or medication affecting the liver and/or the immune system led to exclusion from the studies, which strengthens this review. On the other hand, two limitations are the non-identical study protocols and different laboratory methods used. All the studies used well-established laboratory methods, and although most approaches were similar, differences in methods to measure T cell activity (e.g., chosen lymphoproliferative assay or HBV peptides), in samples used (PBMCs vs. liver samples), in reagents, and in the focus on HBV-specific T cell responses versus global T cell responses in the different studies might account for some of the discrepancies in the results.

## 5. Conclusions

In summary, CTLA-4 is a receptor in the CD28 family of molecules whose activation results in a strong inhibitory signal to T cells. Based on the current systematic review investigating the role of CTLA-4 in the exhaustion state of T cells in CHB patients, it can be concluded that current research on the role of CTLA-4 in T cell exhaustion in patients with CHB is not entirely consistent. However, the current data suggest that CTLA-4 may be a contributing factor in the development of T cell exhaustion and that particularly the upregulation of CTLA-4 on Tregs may be important. The role of CTLA-4 and other co-inhibitory receptors is complex, and further studies are needed to uncover the specific role of the receptor. An interesting path for future research in T cell exhaustion is transcriptome studies of HBV-specific T cells, which can be used to reveal the complete cellular pathways involved in this complex phenomenon instead of investigating single predefined molecules. While immune therapy using CTLA-4 inhibitors can potentially be a promising approach for future treatment of CHB, further research needs to clarify the effect and possible adverse effects, as a major concern is the possible induction of an excessive immune response.

## Figures and Tables

**Figure 1 viruses-15-01141-f001:**
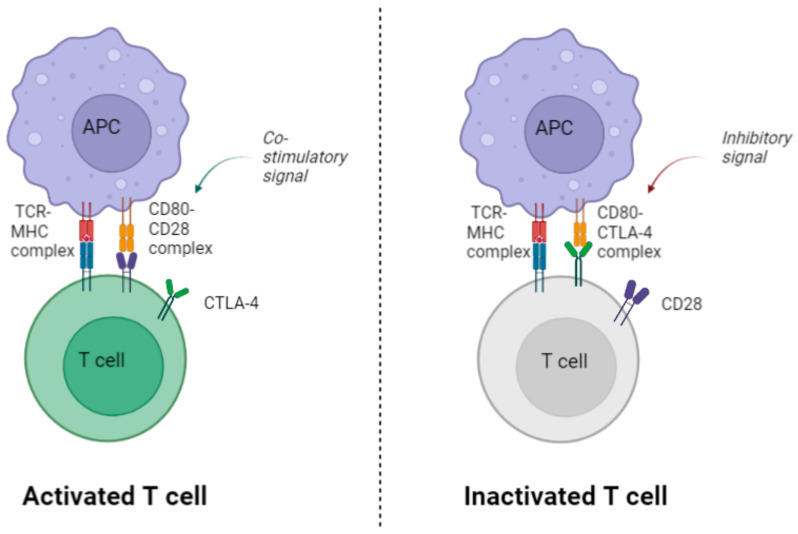
Illustration of the role of CTLA-4 in T cell inactivation. Binding of CTLA-4 to CD80 inhibits the positive co-stimulatory signaling of CD28, resulting in an inhibitory signal in the T cells. CD: Cluster of Differentiation; TCR: T cell receptor; MHC: major histocompatibility complex. Created with Biorender.com.

**Figure 2 viruses-15-01141-f002:**
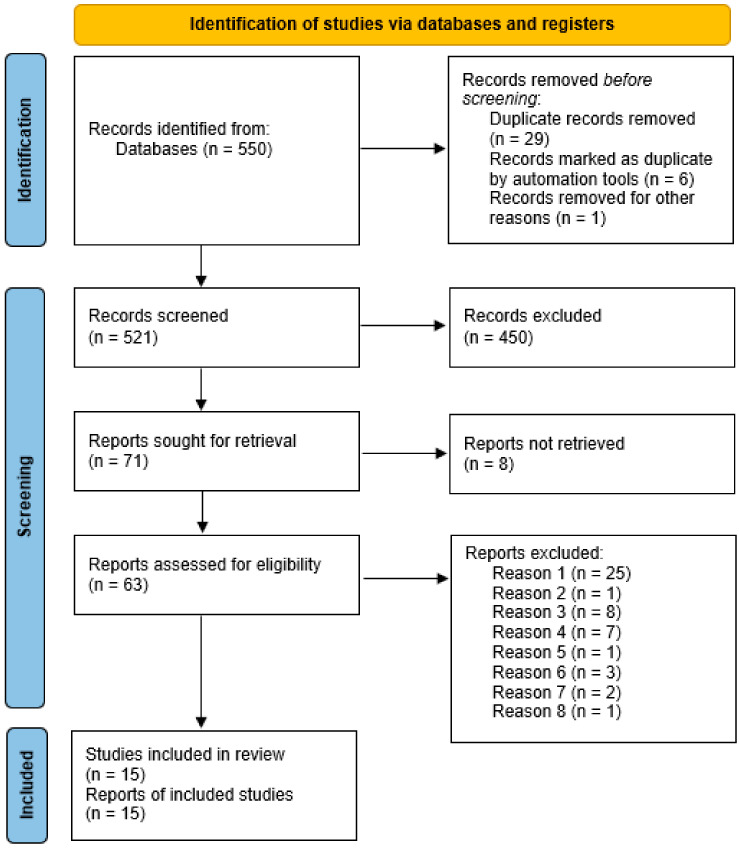
PRISMA 2020 flow diagram of the studies included in the systematic review. Reason 1: all included patients not diagnosed with CHB; reason 2: not human, adult population; reason 3: not an eligible outcome; reason 4: review article or meta-analysis; reason 5: commentary without original results; reason 6: the intervention is not a focus of the study; reason 7: only a subgroup of CHB clinical stage studied; reason 8: full report not in English language. (Page et al. 2021) [20].

**Table 1 viruses-15-01141-t001:** Population Intervention Comparison Outcome (PICO) questions.

	Population	Intervention	Comparison	Outcome
	CHB patients	Hepatitis B virus	Control group	Up- and downregulation of CTLA-4 on T cells Effect on T cell function and exhaustion
Inclusion criteria	Adults ≥ 18 years Human studies	HBsAg-positive > 6 months	Healthy controls Other controls	CD8^+^ T cells CD4^+^ T cells incl. Tregs
Exclusion criteria	Pediatric patients Animal studies Review or meta-analysis Only subgroup of CHB clinical stage studied			Other cell types are main study object

CHB = chronic hepatitis B; CTLA-4 = cytotoxic T lymphocyte associated antigen 4; HBV = hepatitis B virus; Tregs = regulatory T cells.

**Table 2 viruses-15-01141-t002:** Baseline characteristics of included studies.

Authors	Patients (n)	Healthy Controls (n)	Reference Year	Country
Bengsch, B et al. [21]	98	0	2014	Germany
Jiang, D et al. [22]	31	23	2022	China
Park, J. J. et al. [23]	200	20	2016	USA
Peng, G. et al. [15]	79	20	2008	China
Peng, G. et al. [24]	103	30	2011	China
Raziourrouh, B. et al. [25]	66	5	2014	Germany
Schurich, A. et al. [26]	86	23	2011	England
Stoop, J.N. et al. [27]	32	0	2008	Netherlands
Stoop, J.N. et al. [17]	50	23	2005	Netherlands
Tang, Z.S. et al. [28]	52	26	2016	China
Wang, L. et al. [29]	62	30	2014	China
Wang, X. et al. [9]	40	40	2019	China
Wen, C. et al. [26]	134 *	25	2023	China
Yu, Y. et al. [11]	120	0	2009	China
Zhang, H. H. et al. [16]	15	25	2010	China

* = 106 treatment-naive CHB patients, 13 patients with HBV-related hepatic failure, 15 CHB patients in longitudinal cohort.

## Data Availability

This systematic review is registered in PROSPERO (https://www.crd.york.ac.uk/prospero/) accessed on 23 March 2023 with ID number CRD42023404696.

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
