# Peer review of "The Role of CTLA-4 in T Cell Exhaustion in Chronic Hepatitis B Virus Infection"

_viruses, 2023, doi:10.3390/v15051141_

Round 1
Reviewer 1 Report
The systematic review of Apol et al. titled “The role of CTLA-4 in T cell exhaustion in chronic hepatitis B virus infection” summarizes research on CTLA-4 expression on lymphocyte populations during chronic hepatitis B infection. In general, the topic of the review is highly relevant and timely, mostly because T-cell exhaustion in CHB is not fully characterized and treatment efficacy is currently unsatisfactory. Thus CTLA-4 inhibitors may be considered as potentially novel treatment strategy.
Strong points are logical structural division (separate description of each cell subset, description not only of a CTLA-4 phenotype but also, when possible, how it affected lymphocyte function). Importantly, the effects of CTLA-4 blockade or treatment on lymphocyte populations are also provided. A substantial effort has been made not only to describe characteristics of peripheral lymphocyte subsets, but also, where possible, intrahepatic cells, including their distinct specificities. I have several minor points to be addressed:
General comments:
1) The introduction part is too long and too general and could be shortened with focus on HBV-specific immunity, including exhaustion and CTLA-4-related mechanisms. In current form, the description of T-cell exhaustion, the underlying phenomenon of CTLA-4 expression is only one sentence (l. 52-55).
2) While reporting results, please pay attention that the expression was lower/higher only when statistically significant. Otherwise it is confusing.
3) IFN-based treatment may itself have effect on CLTLA-4 expression since it is an immunomodulatory drug. Thus, each time describing the treatment effect on CTLA-4 expression please specify whether it was IFN or NA.
4) Please make result subtitles more informative – summarizing the finding.
5) Please specify each time whether the study concerns peripheral or intrahepatic cells.
6) Please add few words of comment on other iRs different than PD-1, e.g., Tim-3, Lag-3, etc. on lymphocytes in CHB – were those studied?
7) Discussion seems too long and commonly is a repetition of results, with little commenting. Could be re-structured to be more synthetic and concluding.
Specific comments:
1) line 53-54, 122: the sentences are confusing – especially the word “depletion” is inappropriate – it is actually exhaustion
2) for better readability, please provide “+” and “-“ in upper index, e.g., CD4+, CXCR5-
3) The word “reactivated ” could be replaced by “restituted”.
4) lines 470-471: was the association/correlation positive or negative?
5) Lines 509-511: the sentence is unclear – “a subpopulation within a T cell subset is actually the driving force of T cell exhaustion”
6) 536: The word “overwhelming” could be replaced by “excessive”.
Minor editing of English language required
Reviewer 2 Report
In this manuscript, the authors summarized the current progress on the role of CTLA-4 in the development of T cell exhaustion in chronic hepatitis B and showed that the role of CTLA-4 in T cell exhaustion in CHB remain unclear since the evidence in this field is still limited. Although the topic is interesting, the formation and writing of this manuscript is needed to be polished. A review should be concise, and this manuscript looks like a research article not a review. The authors need to interpret the articles rather than combine all the data. I believed that the authors need to reorganize the manuscript since the quality of this manuscript can be better.
The quality of language is good.
